# β-Cyclocitral: Emerging Bioactive Compound in Plants

**DOI:** 10.3390/molecules27206845

**Published:** 2022-10-13

**Authors:** Mohammad Faizan, Sadia Haque Tonny, Shadma Afzal, Zeba Farooqui, Pravej Alam, S. Maqbool Ahmed, Fangyuan Yu, Shamsul Hayat

**Affiliations:** 1Botany Section, School of Sciences, Maulana Azad National Urdu University, Hyderabad 500032, India; 2Faculty of Agriculture, Bangladesh Agricultural University, Mymensingh 2202, Bangladesh; 3Department of Biotechnology, Motilal Nehru National Institute of Technology Allahabad, Prayagraj 211004, India; 4College of Pharmacy, University of Houston, Houston, TX 77204, USA; 5Department of Biology, College of Science and Humanities, Prince Sattam Bin Abdulaziz University, Alkharj 11942, Saudi Arabia; 6Collaborative Innovation Center of Sustainable Forestry in Southern China, College of Forest Science, Nanjing Forestry University, Nanjing 210037, China; 7Department of Botany, Faculty of Life Sciences, Aligarh Muslim University, Aligarh 202002, India

**Keywords:** apocarotenoid, abiotic stress, β-carotene, gene expression

## Abstract

β-cyclocitral (βCC), a main apocarotenoid of β-carotene, increases plants’ resistance against stresses. It has recently appeared as a novel bioactive composite in a variety of organisms from plants to animals. In plants, βCC marked as stress signals that accrue under adverse ecological conditions. βCC regulates nuclear gene expression through several signaling pathways, leading to stress tolerance. In this review, an attempt has been made to summarize the recent findings of the potential role of βCC. We emphasize the βCC biosynthesis, signaling, and involvement in the regulation of abiotic stresses. From this review, it is clear that discussing compound has great potential against abiotic stress tolerance and be used as photosynthetic rate enhancer. In conclusion, this review establishes a significant reference base for future research.

## 1. Introduction

Plants have evolved several types of secondary metabolites as a defensive shield to protect themselves from phytophagous herbivores [1]. Miscellaneous bioactive compounds induce fuel in various beneficial activities such as wound healing and antifungal, anti-inflammatory, and antimicrobial effects. For eco-friendly and substantial environments, the use of biomolecules is incrementally increasing [2]. β-cyclocitral (βCC) 2,6,6-trimethyl-1-cyclohexene-1-carboxaldehyde is an endogenous volatile compound which is derived from the carotenoid β-carotene [3]. Plant protection proficiency from injury of plants due to free radicals, stimulation of enzymes, and extinguishing singlet oxygen function is controlled by bioactive compounds [4]. The functional characterization of carotenoid compounds like hormones, signals, and biosynthesis through the non-enzymatic method is observed in apocarotenoid [5].

Carotenoids are tetraterpene pigments, which extensively distributed in foodstuffs that have always been part of the human diet. Few carotenoids can be converted into retinoids exhibiting vitamin A properties, which is important for humans. Moreover, they are very versatile as they are present in food not only as a vitamin A component, but also as natural pigment, antioxidant, and health curing compound [6]. They play an important role in lethal reactive oxygen species (ROS) scavenging and also actively participate in light absorption and the protection of photosynthetic supplies and functions [7]. The β-ring holding carotenoids like apocarotenoid obtained from carotenoid cleavage dioxygenases-4 (CCD-4 clade), which generate volatile apocarotenoids are known as βCC. It is derived from singlet oxygen ^1^O_2_ invasion, and is present in higher amounts during the chemical reactions in photosynthesis performing cells. Poly-unsaturated fatty acid oxidation is found to function as a co-substrate for performing enzymatic activities [8,9]. Interestingly, βCC contain vital apocarotenoids of β-carotene [10] that play important roles under abiotic stress conditions [11,12]. Plant genomic effect, apoptosis, gene activation processes, and transcription activation are encountered due to low RES and excess ROS oxidation [13,14,15]. Carotenoid degradation involves βCC, β-ionone, β-ionone rendered in water odor. Organoleptic properties (flowery flavor or fragrance) of βCC in beverages, pharmacy and industries are also economically beneficial [16,17]. In many plant species, βCC has been found in leaves, flowers, fruits and roots [18,19]. Moreover, a substantial source of βCC is found in lichens and mosses [20,21].

Although it acts as a powerful repellent and a signal of poor quality food to grazers, such as Daphnia [22] and the cell dissolution of *Microcystis* [23], βCC is beneficial to vascular plant growth. One of the major benefits of βCC is that it plays an important role in growth regulation, enhancing branching, the emergence of lateral roots, and cell division [24]. The production of provitamin carotenoid is highly dependent on βCC, which helps to improve the yield of the bioreactor and growth index [25]. The quantity of βCC is proportional to the amount of LOX 13S- lipoxygenase process which influences the decrease of apocarotenoid in *Solanum lycopersicum* and *A. thaliana* mutants in comparison with the wild type [26]. The deterioration of aquatic and marine organisms in freshwater due to carotenoid compound degeneration lead to the formation of malodor. One of the major pretenses of this odor and taste compound is βCC along with dimethyl sulfide, β-ionone, etc [27]. A number of vegetables and fruits are the sources of violet and raspberry type fragrance in carotenoid compounds.

βCC is capable of eliciting multiple stress signals and gives strength to survive under unfavorable conditions. Stress signaling of βCC is induced by oxidation and degradation of chloroplasts which enhances the production of ROS and the activity of auxin and brassinosteroids. βCC is able to promote root growth in rice, reducing the toxicity of salinity to rice seedlings [24]. Recent studies have found that βCC can enhance the tolerance of *Arabidopsis*
*thaliana* to high light stress [10]. The acclimatization of photo-oxidative stress causes the inhibition of signal modulation of ^1^O_2_ through methylene blue sensitivity, which acts as a zinc finger protein. This function was directed by transcriptome reprogramming [28]. Detoxification activities for resisting oxidative cell damage are increased by βCC [10]. The βCC exhaustion is influenced by foliose lichen during heat and wounding stress [29]. It acts as a secondary precursor for stress signal relocation in *A. *thaliana**. Analogous activities of βCC were also found to occur in dihydroactinidiolide during ^1^O_2_ oxidation [10]. The excess amount of light interference persuades the increasing amount of glycosylated βCC, which immobilizes the signaling molecule [30,31]. As a consequence, drought tolerance is achieved by signaling and pathway activation by using βCC exploitation. The exogenous application of βCC also ameliorates the drought and light stress in *A. thaliana*, Viola tricolor and Capsicum sp. [31]. Antioxidant signaling and crosstalk are imitated during stress, which causes the upregulation of various enzymes like superoxide dismutases (SOD), catalase (CAT) and peroxidase (POD) [32]. Inhibiting traits of βCC on hydroxylase enzymes causes multiple loop-hole creations for establishing a homeostasis pathway. The inter-connectedness between metabolic activities in primary and secondary metabolism is observed which agitate the consistent state of miscellaneous metabolites [33]. Previous studies revealed that βCC is a volatile compound derived from β-carotene oxidation, which mediates the response of cells to singlet oxygen stress. Beside these well-known examples, the latest research unraveled novel apocarotenoid growth regulators and suggested the presence of yet unidentified ones. However, knowledge of βCC involvement in the complex stress signaling network is very limited. This review highlights the structure and functions of βCC in plants. It presents the βCC mediated stress mitigation, as well as signaling cascades in plants. The authors also demonstrated the βCC-mediated involvement in the regulation of the stress response.

## 2. Biosynthesis of βCC

The formation of βCC occurs either by direct oxidation of β-carotene through ROS (^1^O_2_) or by an enzymatic pathway. A family of non-heme iron-dependent enzymes in plants catalyzes the carotenoids by an enzymatic cleavage via 9-cis-epoxycarotenoid cleavage dioxygenases (NCEDs) and carotenoid cleavage dioxygenases (CCDs), resulting in apocarotenoids, an oxidation product [34,35]. The first step in abscisic acid (ABA) production is catalyzed by NCED enzymes cleaving the 11, 12 (11′, 12′) double bond of 9-cis-violaxanthin or 9-cisneoxanthin [36]. Furthermore, CCD enzymes and NCED enzymes do not share cleavage specificities. In *Arabidopsis*, there are four CCDs (CCD1, CCD4, CCD7, and CCD8). It is unknown whether one of these CCDs creates βCC from carotene in plant leaves. In each of the four CCDs in *Arabidopsis* deficient mutants, the accumulation of βCC was not affected, which suggests that β-carotene oxidation mediated by CCD in this species is not a major source of this apocarotenoid [37], despite the fact that between 4 CCDs functional redundancy cannot be ruled out. This is similar in cyanobacteria, where βCC formation aided by CCD was not found [38]. Unlike CCDs that are plastidial, cytosolic CCD1 cleaves the double bonds of 9, 10 (9′, 10′) to produce varying volatiles and apocarotenoids of extensive acyclic or monocyclic apocarotenoids and carotenoids.

The strigolactones biosynthesis is dependent on CCD8 and CCD7 [39]. Since CCD4 has a specific cleavage activity at 9, 10 (9′, 10′) and 5, 6 (5′, 6′) double bond, it does not generate βCC [34,35]. Furthermore, in high light conditions, CCD4 is highly downregulated, which activates the accumulation of βCC [37]. However, the cleavage of β-carotene in citrus from the location 7, 8 (7′, 8′), CCD4b is reported under CCD4 enzyme, which results in the production of βCC [8]. Similarly, another CCD4c in the *Crocus stigma* from CCD4 can cleave carotenoids at 9-10 (9′, 10′), resulting into β-ionone and produces βCC with low efficiency at 7 and 8 (7′, 8′) [40]. For the production of βCC, CCD4b gene in *Vitis vinifera* in the carotenoid-accumulating yeast strain is also reported [41]. Another way for the oxidation of carotenoids can be provided by lipoxygenase [9]. Similarly, in leaves of *Solanum lycopersicum* and *Arabidopsis*, knockout mutants for 13-lipoxygenase LOX2 were reported to have low levels of βCC [26]. On the other hand, in the βCC accumulation under high light and ^1^O_2_ stresses, it is unknown if this enzyme is involved despite the fact that LOX2 is induced under these circumstances [37]. Eventually, from the fungus *Lepista irina,* extracellular fluid purified a peroxidase which produces βCC and other unstable apocarotenoids from the cleavage of β-carotene [42].

When compared to photosystem II, it is thought that photosystem I does not produce considerable amounts of ^1^O_2_. Auto-oxidation of β-carotene can also produce βCC, especially when attacked by the reactive specie ^1^O_2_ [31]. Carotenoids quench ^1^O_2_ through a physical mechanism that involves energy transfer from ^1^O_2_ to the carotenoid, followed by the excited quencher’s thermal decay [43]. However, carotene can be oxidised by ^1^O_2_, allowing ^1^O_2_ to be chemically quenched. ^1^O_2_ is an electrophilic molecule that has a strong affinity for double bonds in carotenoid molecules, oxidizing them and creating a range of apocarotenoids, including βCC [10]. In microalgae, the principal oxidation products of β-carotene are β-ionone and βCC, which release large amounts of these chemicals during summer blooms [44].

## 3. Derivatives of βCC

Recently, a new bioactive compound, βCC, has emerged in a variety of living organisms varying from plants and cyanobacteria to animals and fungi. It is a volatile compound consisting of a short chain of apocarotenoids produced by the non-enzymatic and enzymatic oxidation of the β-carotene. Derivatives of βCC such as lactones, β-cyclocitric acid, glycosylated βCC, 2,2,6-trimethyl-cyclohexanone and 2,6-dimethyl-cyclohexanol are briefly discussed. Lactone, being a potent biologically active compound, occurs naturally and possesses anticancer, antiplasmodial, antifungal and antimicrobial effects. Figure 1 showed the schematic synthesis of lactone. In a previous study, lactone synthesis stated by the addition of NaBH_4_ in the presence of H_2_O/CH_3_OH in βCC (10 g, 0.06 mol) which yielded an alcoholic compound βCC (9.65 g, yield 97%) [45]. This alcohol was treated with CH_3_C(OC_2_H_5_)_3_, CH_3_CH_2_COOH at 137 °C and resulted in (1,3,3-Trimethyl-2-methylene-cyclohex-1-yl) acetic acid ethyl ester. In the presence of KOH/C_2_H_5_OH, this acetic acid ethyl ester was transformed into (1,3,3-Trimethyl-2-methylene-cyclohexyl) acetic acid [46]. Furthermore, this acetic acid was converted to either 1-Bromomethyl-2,2,6-trimethyl-9-oxabicyclo [4.3.0] nonan-8-one with the reagent NBS/THF, CH_3_COOH or to 1-Chloromethyl-2,2,6-trimethyl-9-oxabicyclo [4.3.0] nonan-8-one with the help of NCS/THF, CH_3_COOH. It can also be converted to 1-Iodomethyl-2,2,6-trimethyl-9-oxabicyclo [4.3.0] nonan-8-one with the help of I_2_/KI, NaHCO_3_ which can be further converted to 1,2,2,6-Tetramethyl-9-oxabicyclo [4.3.0] nonan-8-one in the presence of n-Bu_3_SnH. The product 1,2,2,6-Tetramethyl-9-oxabicyclo [4.3.0] nonan-8-one, 1-Iodomethyl-2,2,6-trimethyl-9-oxabicyclo [4.3.0] nonan-8-one, 1-Bromomethyl-2,2,6-trimethyl-9-oxabicyclo [4.3.0] nonan-8-one and 1-Chloromethyl-2,2,6-trimethyl-9-oxabicyclo [4.3.0] nonan-8-one are all lactones derived from βCC [46].

In the leaves of *Arabidopsis*, the oxidation of βCC results in β-cyclocitric acid (2,2,6-trimethyl cyclohexene-1-carboxylic acid) which is responsible for the accumulation under strain conditions [31]. The level of β-cyclocitric acid in plants under drought stress increased by an aspect of 15 and doubled the content of βCC only. An accumulation of β-cyclocitric acid in plants exposed to volatile βCC was observed [31]. In the leaves and fruits of tomato, the levels of β-cyclocitric acid are remarkably more than that of βCC. Unlike in water, the conversion of βCC to β-cyclocitric acid was much faster in plants, suggesting the role of enzymatic catalysis [47]. As it remains undiscovered, there might be a connection of a Baeyer-Villiger monooxygenases that yields esters from carboxylic acids and ketones from aldehydes [48]. In brassinosteroid biosynthesis, this type of enzyme responsible for the oxidation of castasterone to brassinolide has been reported [49]. A regulatory mechanism that modulates the βCC-mediated signaling could be represented by the process of glycosylation of βCC. However, from the total βCC pool, glycosylated βCC was found to represent only a small portion (<2%) [30]. The βCC converts to various compounds like 2,2,6-trimethyl-cyclohexanone and 2,6-dimethyl-cyclohexanol after UV-light exposure [50]. In glycosylated form, βCC can occur in plants, and several glycosyl transferases are induced by the βCC treatment [10].

## 4. Signalling of βCC in Plants

The βCC has emerged as a new dimension for acting as a stress tolerant molecule in adverse conditions. The signaling pathway has been disclosed along with the transportation function within plants. The βCC performs in a hormone-induced marker line and the corresponding mutant responds to phytohormone pathway signals like auxin and brassinosteroids (BRs) and eventually results in cell enlargement [51]. Enzymatic action produces CCD4b from the genetic variance of *Crocus sativus* through β-carotene cleavage in the model plant *Arabidopsis,* which hastens to reduce the dehydration, salinity and oxidation rate [40,52]. The endo-metabolic substances in vascular plants took part in the xenobiotic response with diverse detoxifying agents, such as SCL14, ANAC102, ANAC001 and ANAC031 for oxidation resistance [11]. The βCC induced plants build interdependence with PAP signaling and down-regulate carotenoid substances while ST2A acts as a sulfate donor, and SAL1 has a deleterious effect during the plethora of light and drought stress [53,54]. Along with PAP, Methylerythritol cyclodiphosphate (MEcPP) substrates are also redox regularized and trigger the augmentation rate of the ROS level [55,56,57]. Sustaining photosensitivity during oxidative stress environment in mbs1 mutant crops for signaling pathway, procurement of protein and partial replacement in the nuclei occurs [28].

## 5. Functions of βCC in Plants

The βCC is a volatile organic compound that has been reported to have multiple functions in non-vascular plants (Figure 2). Microalgae discharge βCC, which is responsible for transferring stress signals to homogeneous algae and inducing defence. The former compound plays an allelopathic function on heterogeneous algae and aquatic macrophyte for opposite nutrients, as well as providing protection in opposition to predators [58]. The βCC has been reported to make cell rupture in *Nitzschia palea*, a diatom [59]. Ikawa et al. (2001) [60] reported that in cyanobacterium *Microcystis,* βCC is one of the main emitted volatile organic compounds. Sun et al. (2020) [61] suggested that the toxicity of βCC to cells might be associated with nuclear variation, DNA laddering, caspase 9- and caspase 3-like performance, signifying the initiation of a programmed cell death mechanism. In the case of cyanobacterial bloom, β-cyclocitric acid is produced by the oxidation of the βCC compound by *Microcystis,* indirectly causing toxicity. The production of this acidic compound leads to water acidification, causing chlorophyll loss, cell lysis, and phycocyanin pigment release, resulting in a characteristic blue colour [21]. These studies suggest that βCC and other volatile apocarotenoids are the principal allelopathic agents in cyanobacterial volatile organic compounds, but that at high concentrations, these compounds may be harmful to the emitters. However, no evidence has been found that low levels of βCC can elicit defence mechanisms in photosynthetic bacteria, such as those found in vascular plants. Mosses have also been observed to release volatile chemical compounds that could be used in interspecies communication [62]. Experimental evidence exhibited that photosynthetic activators and enzymatic variance treated with βCC in plants increased the photosynthetic rate, root-shoot expansion and carbon assimilation [63].

The mosses *Hamatocaulis vernicosus* and *Sphagnum fexuosum* compete for the release of volatile organic compounds via increasing emission of a molecule called methyl 2,6, 6-trimethyl-1-cyclohexene-1-carboxylate, which is chemically linked to βCC. As a result, an alarmed mechanism could be set off, signalling the competitive strength of their neighboring moss species. The former is an enzyme that can convert carotene to βCC. The βCC is an intermediary in the ^1^O_2_ signalling pathway, which controls gene expression reprogramming. It eventually causes plant cells to shift from active growth to cellular defence, resulting in stress and adaptation. The bulk of the downregulated gene encoded proteins involved in the development, growth, and biogenesis of cellular components [10]. Upregulated genes, on the other hand, were linked to environmental interactions, stress responses, and cellular mobility. Under normal or light stress conditions, βCC produces a tiny zinc finger protein (MBS1; methylene blue sensitivity 1) that is needed for the proper expression of ^1^O_2_-responsive genes [64]. The βCC is said to have increased the former protein levels while also causing the protein to relocalize to the nucleus [28]. Further research revealed that the *Arabidopsis* mbs1 mutant (deficient in MBS1) was insensitive to βCC and therefore lacked an increase in photo-tolerance after treatment with βCC [28]. MBS1 is thought to be downstream of βCC in the ^1^O_2_ signalling pathway, although it’s precise function remains uncertain. Exogenous reactive substances are inactivated by typical detoxifying enzymes in vascular plants, which remove these molecules in three phases. In the first stage, side groups are introduced or modified in harmful substances such as herbicides, pollutants, and so on. The modified molecule is conjugated to sugar moieties or glutathione in the second stage. Finally, inactivated chemical compounds are compartmentalized [65]. The transcriptome of βCC-treated *Arabidopsis* plants showed activation of detoxification pathways [66]. Several glutathione-S-transferases (GST) and UDP-glycosyltransferases (UDP-glycosyltransferases) were involved in the xenobiotic detoxification process. The GRAS protein (SCL14; SCARECROW LIKE 14) and the glutaredoxin (GRX480/ROXY19) fight for interaction with TGAII transcription factors and mediate the activation/inhibition of a large number of detoxifying enzymes during the modification phase [67]. SCL14-controlled xenobiotic detoxification was induced by βCC and photooxidative stress conditions. Similarly, the scl14 knockout mutant did not respond to βCC and remained susceptible to high light stress even after treatment with βCC [68]. A few signalling cascade components downstream of βCC have been identified. The SCL14/TGA II complex, in particular, modulates the transcriptional levels of a transcription factor that regulates other downstream transcription regulators and, eventually, the redox enzymes of the first phase of the detoxification response [68], thus improving cellular detoxification capabilities. Surprisingly, the loss of MBS1 had no effect on βCC-induced cellular detoxification, indicating that there are two pathways in βCC signalling, one regulated by SCL14 and the other dependent on MBS1 [68]. Lipid peroxidation produces the chemicals that are characteristic of photooxidation and contributes unwaveringly to its toxicity. They decompose into reactive aldehydes (acrolein or 4-hydroxynonenal), which obstruct macromolecule function and cause cell death [69]. Diversified functions of βCC in plants are given in Table 1.

In plants including *Solanum lycopersicum*, *Piper nigrum*, and *Arabidopsis thaliana*, chemical βCC can cause changes in gene expression and promote drought tolerance [70]. The available literature on βCC shows that it may activate a signalling cascade that has yet to be fully described. Dickinson et al. (2019) [24] found that *Arabidopsis* seedlings grown in Petri plates treated with βCC stimulated the growth of primary roots. Increased root lengths may be beneficial under salt and water stress conditions, allowing for better soil exploration and water uptake. βCC has the ability to influence root development in *Solanum lycopersicum* and *Oryza sativa* without relying on auxin or brassinosteroid signalling. However, it is unclear whether βCC causes root growth directly through cell division and elongation or indirectly through the activation of cellular detoxification and resistance to oxidative stress. The molecular processes underlying βCC control of root development will need to be clarified in future research [31].

**Table 1 molecules-27-06845-t001:** Various functions of βCC in Plants.

Plant Species	Functions	References
* 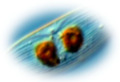 * *Nitzschia palea*	Cell rupture at 0.1–0.5 mg mL^−1^ dose, cyanobacterial cell degradation, change in water color	[59]
* 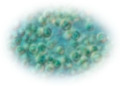 * *Cyanidium caldarium*	Unpalatable water odor	[71]
* 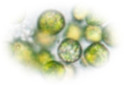 * *Chlorella pyrenoidosa*	Inhibition of cell growth and development	[60]
* 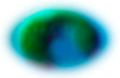 * *Chlamydomonas reinhardtii*	Induce programmed cell death, cause poison to other algae	[61]
* 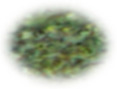 * *Microcystis aeruginosa*	Increase βCC emission, expose high ion concentration	[72]
* 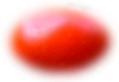 * *Solanum lycopersicum*	Retro nasal olfactory (smell) add to flavor to the fruit, volatile compound induces taste	[73]
* 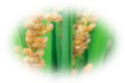 * *Oryza sativa*	Scented rice varieties have aroma, more leaves present in vegetative stage	[74]
* 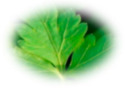 * *Petroselinum crispum*	Helps to produce essential oil and contribute in anti-fungal activity	[75]
* 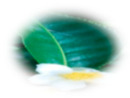 * *Camellia sinensis*	Improve odorant properties and structural functions	[76]
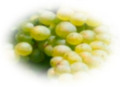 *Grapevines*	Inhibit infestation of spider mite and reduce symptoms	[77]

## 6. βCC Involved in the Regulation of Stress Response

The βCC has recently emerged as a unique plant signal in vascular and non-vascular plants that triggers stress tolerance (Figure 3). Ramel et al. (2012) [10] found that βCC increased tolerance to drought stress and high light-induced oxidative stress in *Arabidopsis thaliana* by altering the expression of several nuclear-encoded genes. Overexpression of the Crocus CCD4b gene in *Arabidopsis* confers tolerance to environmental and oxidative stressors, according to Baba et al. (2015) [52]. βCC plays important role in drought stress tolerance. Drought stress hampers photosynthesis, stomatal conductance, and the respiration and transduction system. In plants, leaf senescence, cell expansion, and yield significantly decreased under drought stress [78,79]. ROS provokes drought stress which puts the oxidation process, assimilation and antioxidant systems at risk [80]. The correlation between βCC and ROS during drought merely demonstrated that increase the rate of βCC aid the reduction of β-carotene in *Solanum lycopersicum*. Simulating stress tolerance, SOD and CAT upregulation altered transcription levels in plants during drought stress (Figure 3). The ^1^O_2_ simulate as a signaling particle, attacking lipid peroxidation within stressed plants [81,82]. In *Arabidopsis thaliana*, elevated levels of singlet oxygen-regulated genes (SORGs) are a crucial indicator in the early stages of stress. Dispensation of βCC or dhA protein aid in lowering photoinhibition in comparison with plants grown in a control or natural stress environment βCC acts as a secondary messenger imparting towards the nucleus through methylene blue sensitive [28]. The uncoupling of chlorophyll compound produces ROS accordingly. The plethora of light stress in the environment decrement the pace of the photosynthetic electron transport chain (PETC) immobilizes the photosystem and inhibits photosynthesis [83,84]. Photoinhibition, incompatibility of photochemical reaction and the D1 protein in photodamage system degrade [85,86]. A copious amount of light energy is mechanized with a non-photochemical extinguishing method that stimulates ^1^O_2_ production in stress conditions and ^1^O_2_ mitigation in the dark light [87]. The excessive re-occurrence of mutagenesis took place in the chlorophyll content [88]. Signaling molecules helps to overcome photodegradation in the light stress condition using a wide range of carotenoid content. In high light stress, carotenoid oxidation took place in PS–II which alters the nuclear gene expression [89]. Moreover, low temperature caused the progressive mitigation of βCC content and the substantial increase in H_2_O_2_ in *Solanum lycopersicum* [90,91]. Plant reaction after treatment with βCC under stress condition is described in Table 2.

Salt stress hampers plant growth amplification, ionic balance, nutritional symmetry and stomatal function in shoots [92,93,94]. Water absorption blockage, water potential alleviation, oxidative stress occurrence and programmed cell death (PCD) strike the plants functioning during salinity stress [95]. Executer 1 and 2 proteins are accountable for cell death in each route [82]. Enhanced root ontogenesis illustrates a favorable impact on salinity stress via βCC application. Rice and *Arabidopsis* roots in considerable depth stimulate plant vigor and genomic functions which help to defeat stress through nuclear transportation and ion activation [24]. In the presence of carotenoid compounds, the *S. lycopersicum* crop demonstrated enhanced oxidation and biochemical synthesis [31]. The inhibition of curly leaves, prevention of leaves wilting, ameliorating relative water content (RWC) and the stomatal opening is accelerated in βCC treated plants [7]. However, the functioning mechanism of βCC in plants has yet to be extensively studied in both long and short-duration experiments. Photo-oxidative stress is a consecutive production of ROS in a pattern which is harmful to the antioxidant defense system. As a result, the photo-oxidation process continues to develop toward a pessimistic function, hazardous chemicals get acclimated, and chloroplast damage happens and eventually yields losses [96]. Photo-oxidation took place in leaves, ontogeny led to senescence and oxidative cascade occurred under stressed conditions [97,98]. The abundance of stress-tolerant genes becomes active in the raised ROS [99,100,101]. In the presence of elevated ROS, singlet oxygen oxidizes promptly in the environment and occurs in PS-II complexes, enabling photochemical quenching of ^3^Chl* transfer [102]. ROS is responsible for leaf damage, mitigating photosynthesis competence, photoproduction and photo-oxidative injury [6,70,103,104,105]. Lichens and cyanobacterial compounds contain reactive radicals in the physiologically active stress condition [29].

**Table 2 molecules-27-06845-t002:** βCC mediated abiotic alteration in plants.

Plants Name	Stress	Effect in Plants	References
*Arabidopsis thaliana*	Light stress	βCC acts as a secondary messenger, alters transcription of singlet oxygen via MBS protein	[89]
*Solanum lycopersicum*	Drought stress	Increases the osmolyte accumulation, root and branching, superoxide exclusion and improves stress tolerance	[7]
*Arabidopsis thaliana*	Photo-oxidative stress	ROS production, changes gene expression	[10]
*Viola tricolor*	Drought stress	Elicits drought stress, no wilting occurs in water lack condition	[31]
*Arabidopsis thaliana*	Oxidative stress	Influences gene expression, suppression in transgenic plants, Cu/Zn SOD is induced	[106]
*Oryza sativa*	Salt stress	Tolerates the adverse salinity stress condition and increased vigor of plants	[24]
*Arabidopsis thaliana*	Water stress	Overlaps genetic responses, showing the impact of singlet oxygen	[107]
*Microcystis cyanobacteria*	Water stress	Oxidation and acidification of βCC; Blue color forms, pH is reduced	[47]
*Arabidopsis thaliana*	Light stress	SCL14 enhances genetic responses, upregulation of lipid peroxidation	[68]
*Oryza sativa*	Salt stress	Promotes cell division in root meristems and stimulates lateral root branching	[24]

## 7. Conclusions and Future Perspective

As an emerging molecule, βCC has been gaining increasing interest to provide stress tolerance due to their excellent features resulting from the altering of the expression of several nuclear-encoded genes. In this work, novel apocarotenoids of β-carotene and βCC applications for stress tolerance were systematically reviewed. Their biosynthesis, derivatives and signalling were discussed. Furthermore, the promising functions of βCC were also discussed and potential directions for future work were suggested.

## Figures and Tables

**Figure 1 molecules-27-06845-f001:**
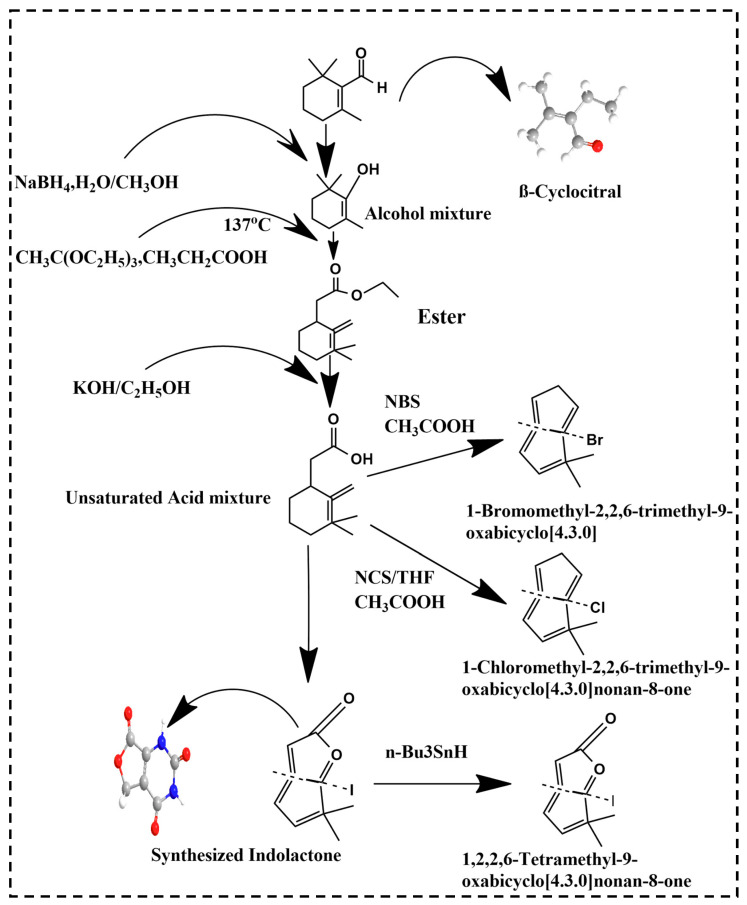
Lactone synthesis from βCC.

**Figure 2 molecules-27-06845-f002:**
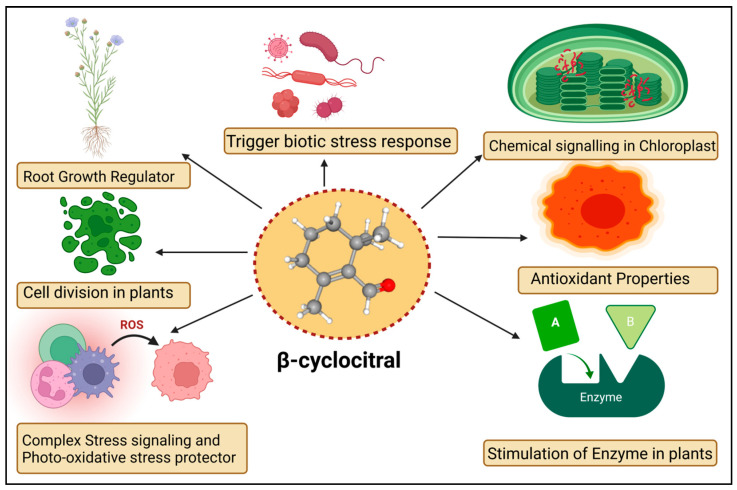
Diagrammatic sketch describing different functions of β-cyclocitral in plants.

**Figure 3 molecules-27-06845-f003:**
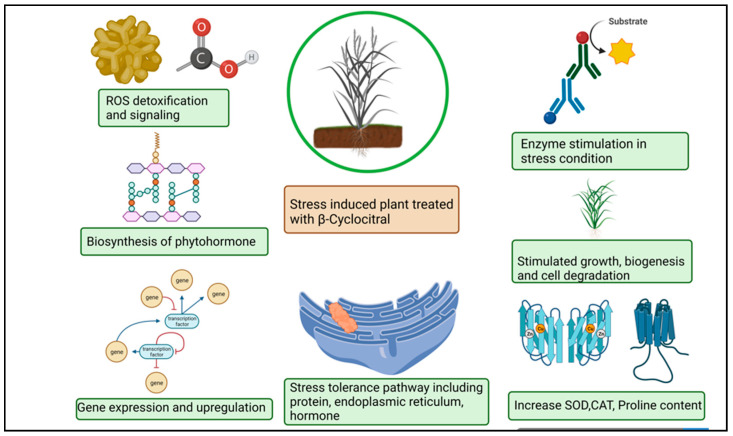
Schematic representation of β-cyclocitral effects on plants under abiotic stress.

## Data Availability

Not applicable.

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
