# Peer review of "β-Cyclocitral: Emerging Bioactive Compound in Plants"

_molecules, 2022, doi:10.3390/molecules27206845_

Round 1

Reviewer 1 Report

The authors describe the potential role of β-cyclocitral, βCC biosynthesis, signaling, and involvement in the regulation of abiotic stresses. However, with minor revisions, this review would be reasonable to accept for publication in Molecules.

1.     Chemical name of β-cyclocitral 2,6,6-trimethyl-1-cyclohexene-1-carboxaldehyde. Provide in the introduction section.

2.     If you provide the Schematic representation about the class of carotenoids and their uses. Please see the reference (Plant Biology 26 (2021) 100203)

3.     Provide the reference  in the introduction section Current Plant Biology 26 (2021) 100203

4.     Page No. 2 Line 88 Replace A. thaliana with Arabidopsis thaliana

5.     Provide the ChemDraw representation for the synthesis of lactones from βCC.

6.     Page No. 4 Line 155 Replace C2H OH with C2H5OH

7.     Provide the page numbers for references 75 and 79.

8.     For ref. 92 Replace O2.−/H2O2 with O2.−/H2O2

Author Response

Author responses to reviewer (1) comments

(Molecules-1931935 R1)

To Reviewer #1

Remarks: The authors describe the potential role of β-cyclocitral, βCC biosynthesis, signaling, and involvement in the regulation of abiotic stresses. However, with minor revisions, this review would be reasonable to accept for publication in Molecules.

Response: The authors are very thankful to the anonymous Reviewer for the appreciation, valuable suggestions, comments and scientific criticism of manuscript for its further improvement.

Remarks: Chemical name of β-cyclocitral 2,6,6-trimethyl-1-cyclohexene-1-carboxaldehyde. Provide in the introduction section.

Response: Added as per your suggestion.

Remarks: If you provide the Schematic representation about the class of carotenoids and their uses. Please see the reference (Plant Biology 26 (2021) 100203).

Response: Reference added as per your suggestion.

Remarks: Provide the reference in the introduction section Current Plant Biology 26 (2021) 100203.

Response: Reference cited.

Remarks: Page No. 2 Line 88 Replace A. thaliana with Arabidopsis thaliana

Response: Corrected as per your suggestion.

Remarks: Provide the Chem Draw representation for the synthesis of lactones from βCC.

Response: Added figure as per your suggestion as Figure 1 in the text.

Remarks: Page No. 4 Line 155 Replace C2H OH with C2H5OH

Response: Corrected as per your suggestion.

Remarks: Provide the page numbers for references 75 and 79.

Response: Added as per your suggestion.

Remarks: For ref. 92 Replace O2.−/H2O2 with O2.−/H2O2

Response: Corrected as per your suggestion.

All the suggestions and comments of the reviewer have been accepted by the authors and the manuscript has been corrected accordingly. A thorough internal reviews was also performed in the whole MS for possible improvement, changes highlighted in Track Change Format supplied MS. We hope the response meets the reviewer approval.

Reviewer 2 Report

The manuscript titled “β-Cyclocitral: Emerging Bioactive Compound in Plants” described β-Cyclocitral biosynthesis, signaling pathways as well as its biological function. In addition, authors describe the βCC involvement in the regulation of stress response of the plant. Although, I find the topic interesting and appreciate the authors' efforts, but many flaws need to be resolved.
The authors have not mentioned a clear rationale for the study design. It should be very clear with the recent references on what was lacking and what prompted the authors to conduct this study.
In addition, manuscript write up is not up to the mark. It seems there is no coherence of the sentences, which is making reviewer very difficult to understand the theme of the manuscript.

Author Response

Author responses to reviewer (2) comments

(Molecules-1931935 R1)

To Reviewer #2

Remarks: The manuscript titled “β-Cyclocitral: Emerging Bioactive Compound in Plants” described β-Cyclocitral biosynthesis, signaling pathways as well as its biological function. In addition, authors describe the βCC involvement in the regulation of stress response of the plant. Although, I find the topic interesting and appreciate the authors' efforts, but many flaws need to be resolved.

Response: The authors are very thankful to the anonymous Reviewer for the appreciation, valuable suggestions, comments and scientific criticism of manuscript for its further improvement.

Remarks: The authors have not mentioned a clear rationale for the study design. It should be very clear with the recent references on what was lacking and what prompted the authors to conduct this study.
Response: Corrected as per your suggestion.

Remarks: In addition, manuscript write up is not up to the mark. It seems there is no coherence of the sentences, which is making reviewer very difficult to understand the theme of the manuscript.

Response: Corrected as per your suggestion.

All the suggestions and comments of the reviewer have been accepted by the authors and the manuscript has been corrected accordingly. A thorough internal reviews was also performed in the whole MS for possible improvement, changes highlighted in Track Change Format supplied MS. We hope the response meets the reviewer approval.

Reviewer 3 Report

- Line 80: convert uppercase to lowercase for this sentence: METHYLENE BLUE SENSITIVITY

- Line 131: unify the font type

-Line 238-239: convert uppercase to lowercase for this sentence: METHYLENE BLUE SENSITIVITY

- Line 326: uppercase only the first letter for the word: EXECUTER

- The authors need to include a section introducing β-cyclocitral as aroma impact compound in several plants.

- References to be quoted:

1- β-Cyclocitral Does Not Contribute to Singlet Oxygen-Signalling in Algae, but May Down-Regulate Chlorophyll Synthesis; https://doi.org/10.3390/plants11162155

2- Toxic mechanism of two cyanobacterial volatiles β-cyclocitral and β-ionone on the photosynthesis in duckweed by altering gene expression; https://doi.org/10.1016/j.envpol.2022.119711

3- Effects of herbivory on carotenoid biosynthesis and breakdown;

4- Reactive oxygen species in photosystem II: relevance for oxidative signaling; https://doi.org/10.1007/s11120-022-00922-x

5- The role of carotenoids as a source of retrograde signals: impact on plant development and stress responses; https://doi.org/10.1093/jxb/erac292

6- Control of chloroplast degradation and cell death in response to stress; https://doi.org/10.1016/j.tibs.2022.03.010

7- Plant environmental sensing relies on specialized plastids; https://doi.org/10.1093/jxb/erac334

8- Effects of high light and temperature on Microcystis aeruginosa cell growth and β-cyclocitral emission; https://doi.org/10.1016/j.ecoenv.2020.110313

9- ROS-derived lipid peroxidation is prevented in barley leaves during senescence; https://doi.org/10.1111/ppl.13769

10- Elucidation of Phenomena Involving Cyanobacteria in Freshwater Ecosystem by Chemically Ecological Approach;

Coordination of Chloroplast Activity with Plant Growth: Clues Point to TOR; https://doi.org/10.3390/plants11060803

12- The Origin of Teratogenic Retinoids in Cyanobacteria; https://doi.org/10.3390/toxins14090636

13- Plant carotenoids: recent advances and future perspectives; https://doi.org/10.1186/s43897-022-00023-2

14- Transient expression systems to rewire plant carotenoid metabolism; https://doi.org/10.1016/j.pbi.2022.102190

15- β-Cyclocitral is a conserved root growth regulator; https://doi.org/10.1073/pnas.1821445116

16- β-Cyclocitral, a Master Regulator of Multiple Stress-Responsive Genes in Solanum lycopersicum L. Plants; https://doi.org/10.3390/plants10112465

17- METHYLENE BLUE SENSITIVITY 1 (MBS1) is required for acclimation of Arabidopsis to singlet oxygen and acts downstream of β-cyclocitral; https://doi.org/10.1111/pce.12856

Author Response

Author responses to reviewer (3) comments

(Molecules-1931935 R1)

To Reviewer #3

The authors are very thankful to the anonymous Reviewer for the appreciation, valuable suggestions, comments and scientific criticism of manuscript for its further improvement.

Remarks: Line 80: convert uppercase to lowercase for this sentence: METHYLENE BLUE SENSITIVITY

Response: Corrected

Remarks: Line 131: unify the font type

Response: Corrected as per your suggestion.

Remarks: Line 238-239: convert uppercase to lowercase for this sentence: METHYLENE BLUE SENSITIVITY

Response: Corrected as per your suggestion.

Remarks: Line 326: uppercase only the first letter for the word: EXECUTER

Response: Corrected.

Remarks: The authors need to include a section introducing β-cyclocitral as aroma impact compound in several plants.

Response: Added as per your suggestion.

Remarks: References to be quoted

Response: Cited

All the suggestions and comments of the reviewer have been accepted by the authors and the manuscript has been corrected accordingly. A thorough internal reviews was also performed in the whole MS for possible improvement, changes highlighted in Track Change Format supplied MS. We hope the response meets the reviewer approval.

Round 2

Reviewer 2 Report

The manuscript titled “β-Cyclocitral: Emerging Bioactive Compound in Plants” described β-Cyclocitral biosynthesis, signaling pathways as well as its biological function is been revised as per reviewer comments. Here are some remarks that have to be consider in further submission

·         Introduction section- Authors are recommended to write introduction with better clarity of the objective. Introduction looks very un-organize, too many description of different elements, however, I suggest to focus on β-Cyclocitral and its scope of research to be discuss in this manuscript. I feel there is no coherence of sentences, which makes reviewer very difficult to understand the theme.

·         Detailed review is been marked in pdf file, kindly refer to pdf pages of the manuscript.

Author Response

Author responses to reviewer (2) comments

(Molecules-1931935 R2)

To Reviewer #2

Remarks: The manuscript titled “β-Cyclocitral: Emerging Bioactive Compound in Plants” described β-Cyclocitral biosynthesis, signaling pathways as well as its biological function is been revised as per reviewer comments. Here are some remarks that have to be consider in further submission.

Response: The authors are very thankful to the anonymous Reviewer 2 for the appreciation, valuable suggestions, comments and scientific criticism of manuscript for its further improvement.

Remarks: Introduction section- Authors are recommended to write introduction with better clarity of the objective. Introduction looks very un-organize, too many description of different elements, however, I suggest to focus on β-Cyclocitral and its scope of research to be discuss in this manuscript. I feel there is no coherence of sentences, which makes reviewer very difficult to understand the theme.

Response: Corrected as per your suggestions.

Remarks: Detailed review is been marked in pdf file, kindly refer to pdf pages of the manuscript.

Response: Corrected as per your suggestions.

All the suggestions and comments of the reviewer 2 have been accepted by the authors and the manuscript has been corrected accordingly. A thorough internal reviews was also performed in the whole MS for possible improvement, changes highlighted in Track Change Format supplied MS. We hope the response meets the reviewer approval.

Reviewer 3 Report

-

Author Response

Author responses to reviewer (3) comments

(Molecules-1931935 R2)

To Reviewer #3

Remarks: English language and style are fine/minor spell check required.

Response: Corrected and checked.